# Modeling and Optimal Operating Conditions of Hollow Fiber Membrane for CO_2_/CH_4_ Separation

**DOI:** 10.3390/membranes13060557

**Published:** 2023-05-29

**Authors:** Dheyaa J. Jasim, Thamer J. Mohammed, Hamed N. Harharah, Ramzi H. Harharah, Abdelfattah Amari, Mohammed F. Abid

**Affiliations:** 1Department of Petroleum Engineering, Al-Amarah University College, Maysan 62006, Iraq; dhyiaa.joumaa@alamarahuc.edu.iq; 2General Company for Food Products, Ministry of Industry and Minerals, Baghdad 10011, Iraq; 3Chemical Engineering Department, University of Technology, Baghdad 10011, Iraq; thamer.j.mohammed@uotechnology.edu.iq; 4Department of Chemical Engineering, College of Engineering, King Khalid University, Abha 61421, Saudi Arabia; hhharharah@kku.edu.sa (H.N.H.); abdelfattah.amari@enig.rnu.tn (A.A.); 5Research Laboratory of Processes, Energetics, Environment and Electrical Systems, Department of Chemical Engineering and Processes, National School of Engineers, Gabes University, Gabes 6072, Tunisia; 6Department of Oil & Gas Refining Engineering, Al-Turath University College, Baghdad 27134, Iraq; mohammad.fadhil@turath.edu.iq

**Keywords:** CO_2_ capture, asymmetric HFM, flux, glassy polymer, COMSOL

## Abstract

In this work, the capture of carbon dioxide using a dense hollow fiber membrane was studied experimentally and theoretically. The factors affecting the flux and recovery of carbon dioxide were studied using a lab-scale system. Experiments were conducted using a mixture of methane and carbon dioxide to simulate natural gas. The effect of changing the CO_2_ concentration from 2 to 10 mol%, the feed pressure from 2.5 to 7.5 bar, and the feed temperature from 20 to 40 °C, was investigated. Depending on the solution diffusion mechanism, coupled with the Dual sorption model, a comprehensive model was implemented to predict the CO_2_ flux through the membrane, based on resistance in the series model. Subsequently, a 2D axisymmetric model of a multilayer HFM was proposed to simulate the axial and radial diffusion of carbon dioxide in a membrane. In the three domains of fiber, the CFD technique was used to solve the equations for the transfer of momentum and mass transfer by using the COMSOL 5.6. Modeling results were validated with 27 experiments, and there was a good agreement between the simulation results and the experimental data. The experimental results show the effect of operational factors, such as the fact that temperature was directly on both gas diffusivity and mass transfer coefficient. Meanwhile, the effect of pressure was exactly the opposite, and the concentration of CO_2_ had almost no effect on both the diffusivity and the mass transfer coefficient. In addition, the CO_2_ recovery changed from 9% at a pressure equal to 2.5 bar, temperature equal to 20 °C, and a concentration of CO_2_ equal to 2 mol%, to 30.3% at a pressure equal to 7.5 bar, temperature equal to 30 °C, and concentration of CO_2_ equal 10 mol%; these conditions are the optimal operating point. The results also manifested that the operational factors that directly affect the flux are pressure and CO_2_ concentration, while there was no clear effect of temperature. This modeling offers valuable data about the feasibility studies and economic evaluation of a gas separation unit operation as a helpful unit in the industry.

## 1. Introduction

Solvent absorption, cryogenic fractionation, and adsorption are the traditional methods for separating and purification of gas mixtures. Equipment complexity, energy consumption, and high capital costs do not prevent them from being mature and reliable methods. Therefore, the membrane separation method represents a promising solution for the scientific and industrial community due to low capital costs, ease of design and operation, low operating and maintenance costs, low energy consumption, and the absence of environmental damage [1,2].

The global market for gas separation membranes was estimated at USD 897 million in the year 2022, and is projected to reach a revised size of USD 1.1 billion by 2026 [2]. The porous and non-porous membranes are utilized to detach CO_2_ from natural gas; however, in large-scale applications, all the membranes depend on a high-density polymer membrane. The techniques of gas isolation in this type of membrane rely on a solution-diffusion mechanism [3,4].

Some standards mostly impact the selection of the membrane when utilized for a particular implementation, such as mechanical efficacy at the operative parameters, stability, yield, separation effectiveness, et cetera. Four fundamentals should be accurately tested for the membrane Gas Separation operation:-The material (permeability and separation factors).-The membrane structure and thickness (permeance).-The membrane configuration (e.g., flat and hollow fiber).

Both membrane’s permeability and selectivity influence the economics of a Gas Separation membrane process. Selectivity is a main factor in achieving both high output purity and yield [5,6].

### 1.1. Material and Construction of Gas Membrane

According to the properties of the material, membranes are distributed into polymeric, inorganic, and metallic membranes. Moreover, polymeric membranes have a main part in commercial applications for their distinct economy and competing efficiency. Polymeric membrane substances can be categorized into glassy and rubbery polymers [6]. Polycarbonates (PC), polyimide (PI), polyethersulfone (PESf), Polysulfone (PSF), and cellulose acetate (CA) are popular polymeric membranes. Polysulfone (PSF) is one of the exceedingly examined polymer membrane substances. It is in general utilized to divide gas because of its cheap cost, chemical stability, good strength, high durability to plasticization, and reasonable gas selectivity [7,8].

Polymeric membranes, mostly used for gas segregation, are in general asymmetric or composite and rely on a gas-solution diffusion mechanism. These membranes, manufactured as plane sheets or hollow fibers, have a slim, high-density coat on the microporous prop that affords mechanical vigor. Usually, polymeric membranes give high selectivity compared to porous inorganic substances due to their low free space. Their permeability and selectivity are in negative relation; as permeability is increased, selectivity is decreased, and vice versa. Nowadays, only about nine polymer substances are used for the manufacturing of about 85% of commercially made membranes [9,10].

### 1.2. Membrane Configuration

There are three different models of membranes commonly used in the industrial field, the plate and frame module, spiral wound module, and hollow fiber module. The preference between the three sets counts on the following requests being performed:-High firmness density.-Reasonable distribution of fluid.-Good stability of mechanical, thermal, and chemical properties.-Low-pressure difference.-Low-cost fabrication.-Simplicity in maintenance and running.-The potency of membrane change.-The potential of changing the system size.-The potency of decontamination.

However, as the decontamination capacity is of less significance in gas segregation the major concern of module layout is a good packing density [11,12,13,14]. A hollow fiber module of 0.04 m^3^ can accommodate an active surface area of 575 m^2^, while the same volume of a spiral wound design can only accommodate 30 m^2^ [15].

Asymmetric hollow fibers have tubular shapes and show rising fluxes required for productive segregations due to the ability to reduce the segregation layer to a thin skin on the exterior surface of the membrane [16,17]. Due to the cheap cost, dense material, and high ratio of surface area to volume, hollow fiber membranes are largely used in gas segregation implementations. Additionally, the asymmetric structure of the hollow fibers shows good mechanical strength and reduces the membrane resistance against the transfer of the components [18,19].

The main objective of this study is represented by three steps. The first step included a proposal for a mathematical model based on the resistance in series to mass transfer. In the second step, the effect of operational factors on the diffusivity and mass transfer coefficient on both sides of the membrane, as well as gas solubility, gas permeability, and selectivity for the membrane material, was investigated. Finally, the experiment results were simulated using the CFD model and compared with the mathematical model, and the effect of operational factors on the recovery and flow of carbon dioxide through the membrane was studied.

## 2. Theory and Mathematical Model

Three sub-models are needed to accomplish the characterization of the HFM (Hollow Fiber Membrane) model: two sub-models depict the flow on each side of the membrane, and the third identifies the segregation technique in the membrane and any pored backing material [16]. A mathematical model of CO_2_ captured from a gas mixture using asymmetric HFM was developed by resistance in a series approach for the three domains of the membrane. Figure 1 displays the concentration profiles of CO_2_ on both sides of the membrane, including the effect of the three resistances on the mass transfer. Carbon dioxide transfer across the membrane is governed by Fick’s law in the gas–membrane interfaces, and the thermodynamic equilibrium is existing. In the law of Fick’s, the concentration of carbon dioxide on the surface of the membrane is related to the partial pressure of this gas and is governed by the dual-mode theory.

At a steady-state process, the rate of mass transfer of piercing components, *i*, across the three resistances is as coming after, by suggesting no variation in the area for mass transfer through the membrane:(1)$$N_{i}=k_{iF}\left(C_{i,F}-C_{i,m1}\right),$$
(2)$$N_{i}=\frac{S_{i}D_{Mi}}{l_{M}}\left(C_{i,m1}-C_{i,m2}\right),$$
(3)$$N_{i}=k_{iP}\left(C_{i,m2}-C_{iP}\right).$$

After combining the three equations to eliminate the intermediate concentration, $C_{i,m1}$ and $C_{i,m2}$ (kmol/m^3^), the equation becomes as follows [18]:(4)$$N_{i}=\frac{C_{iF}-C_{iP}}{\frac{1}{k_{iF}}+\frac{l_{M}}{S_{i}D_{Mi}}+\frac{1}{k_{iP}}},$$
where $N_{i}$ is the molar transmembrane flux of species *i* (kmol/s-m^2^), $C_{iF}$ and $C_{iP}$ are concentration on the feed and permeate, respectively (kmol/m^3^), $D_{Mi}$ is the diffusivity of the solute in the membrane (m^2^/s), $S_{i}$ is the solute solubility in the membrane [m^3^(gas)/m^3^ (*membrane*)], $l_{M}$ is the thickness of the membrane (m), and $k_{iF},\;k_{iP}$ are the mass transfer coefficient on the feed and permeate, respectively (m/s).

### 2.1. Physical Properties of the Gas Mixture

The flux of carbon dioxide through the regions of the membrane is calculated from Equation (4), and solving this equation requires many inputs, such as viscosity and diffusivity. For gas mixtures, the dynamic viscosity is calculated according to the semiempirical equation with the following expression [20,21,22]:(5)$$\mu_{mix}=\sum_{i=1}^{n}\frac{\mu_{i}x_{i}}{\sum_{j=1}^{n}x_{j}\emptyset_{ij}},$$
where, $\emptyset_{ij}$ is defined as:(6)$$\emptyset_{ij}=\left(\frac{1}{8^{0.5}}\right){\left(1+\frac{M_{i}}{M_{j}}\right)}^{-0.5}{\left[1+{\left(\frac{\mu_{i}}{\mu_{j}}\right)}^{0.5}{\left(\frac{M_{i}}{M_{j}}\right)}^{0.25}\right]}^{2},$$

In Equations (5) and (6), *n* is the number of species in the mixture, and *x_i_* and *x_j_* are the mole fractions of species *i* and *j*, respectively. Where $M_{i}$ and $M_{j}$ are the molecular weight of species *i* and *j*, respectively (kg/mol), and $\mu_{i}$ and $\mu_{j}$ are the viscosity of pure gases *i* and *j.*

### 2.2. Gas Diffusivity in the Membrane Regions

The diffusivity in the tube and permeate sides are calculated in terms of binary diffusion coefficients from the Chapman–Enskog Equation [22]:(7)Dij=1.8583×10−27T31Mi+1MjPfσij2ΩD−1,

In Equation (7), σ represents the collision diameter of the components, and $\Omega_{D}^{-1}$ is the dimensionless collision integral. In the region of the membrane, the solution-diffusion model is the most widely used transport model for permeation in the polymer membrane. In this mechanism, the gas dissolves in the membrane material and diffuses as a result of the pressure difference on both sides. The separation is achieved between different gases because of the differences in the amount of gas that sorbs and dissolves in the membrane and the rate at which the gas diffuses through the membrane [20].

The sorption step in the glass polymer used here is completely different from the rubbery polymer and cannot be described by the conventional model. Specifically, the NELF model is used to estimate the sorption in glass Polymer membranes [23,24].

Essentially, gas molecules penetrate through the voids in the polymeric chains, causing the gas to diffuse across the membrane. The free volume is adopted in calculating the diffusion coefficient of many gases through different polymeric membranes [25,26].

Prediction of gas diffusion (standard condition) through the polysulfone membrane is made using the Doolittle relation, and the diffusion coefficient of penetrating components into the membrane is linked to the volumetric fraction of the polymer [27,28,29].
(8)$$D_{Mi}^{o}=A\times e^{\left(\frac{-B}{FFV}\right)},$$
where *FFV* is the fractional free volume and *A* (cm^2^ s^−1^) and *B* are constants. The *FFV* is estimated utilizing the Bondi method [23], as follows:(9)$$FFV=\frac{\overset{*}{\overbrace{v}}-{\overbrace{v}}_{o}}{\overset{*}{\overbrace{v}}},$$
where, $\overset{*}{\overbrace{v}}\;and\;{\overbrace{v}}_{o}$ are the specific volume of polymer and the being used specific volume, respectively. The amounts of free volume constants are listed in Table 1.

The Arrhenius expression is used to consider the influence of temperature on the coefficient of gas diffusion in the polymer membrane [28].
(10)$$D_{Mi}=D_{Mi}^{o}e^{\left(\frac{-E_{d}}{RT}\right)},$$
where $E_{d}$ is the apparent activation energy for diffusion (kcal/mol). The amounts of $D_{Mi}^{o}$ and $E_{d}$ for the membrane are listed in Table 2.

### 2.3. Mass Transfer Coefficients

The mass transfer coefficient of the gas phase on the feed side, depending on the fluid hydrodynamic of the tube side, is estimated using the following correlation:(11)Shg=0.662Scg13Reg12,

${Sh}_{g},\;{Sc}_{g},\;and\;{Re}_{g}$ are the Sherwood, Schmidt, and Reynolds numbers, respectively.
(12)$$k_{g}=\frac{{Sh}_{g}D_{ij}}{d},$$
where $D_{ij},$ and *d* is the diffusivity of gas (*i*) in the gas mixture and the diameter of membrane fiber, respectively [29].

On the permeate side, the coefficient of mass transfer relies on the fluid hydrodynamic of the shell-side, which is affected by the packing density and similarity, which is affected by the packing density and similarity of the distance between fibers. From the following relationship, the mass transfer coefficient can be calculated.
(13)$${Sh}_{g}=5.85\left(1-\vartheta \right)\left(\frac{d_{h}}{L}\right){Re}_{g}^{0.6}{Sc}_{g}^{0.33},$$
where ϑ is the packing density, *L* is the membrane length, and $d_{h}$ is the hydraulic diameter [30].

According to the model of Happel’s free surface, only a part of the gas embracing the fiber is counted, which may be considered a circular cross-section. Taking into account the active area and hexagonal form shell unit surrounding every fiber, the hydraulic diameter of the shell unit of each fiber is estimated as [31]:(14)$$A=A_{O}-A_{I},$$
(15)$$P=P_{O}-P_{I},$$
(16)$$d_{h}=\frac{4A}{P}.$$

Figure 2 shows the circumference and cross-sectional area of the gas slice around the outer diameter of the fibers.

Calculating the area of the hexagon was performed by dividing the total cross-sectional area of the membrane by the number of 3800 fibers, after that, finding the length of the side *S*, then finding the perimeter of the hexagon, and calculating the hydraulic diameter of the gas slice on the side of the permeate.

The Denes membrane mass transfer coefficient $\left(k_{m}\right)$ can be calculated using the following equation:(17)$$k_{m}=\frac{D_{Mi}}{\delta },$$
where δ represents the thickness of the membrane layer.

## 3. CFD Simulation Model

A CFD simulation model of CO_2_ captured from the gas mixture using asymmetric HFM is developed by deriving and solving the governing equation for the three domains of the membrane. Several flow sub-models have been developed to account for flow through different module geometries. Figure 3 shows the shell, membrane, and tube domains, as well as the flow pattern and fiber side feed.

The solution-diffusion model has been used to display the mass transfer. The as follows hypotheses are considered in the formulated model:-Steady-state and isothermal conditions.-Fick’s law was used to describe the diffusion mechanism.-Ideal gas behavior.-The Newtonian-type fluid.-Neglecting the support layer (ignoring the resistance).-Two-dimensional flow patterns.-The driving force in the model is the pressure difference.-All fibers have uniform outer and inner diameters.

Based on these assumptions, the main equations governing gas separation in the membrane are derived from the material and momentum balance of the various hollow fiber composite membrane sections.

### 3.1. Material Balance

The continuity equations in various parts of the hollow fiber composite membrane are as follows:

#### 3.1.1. Feed Side (Tube Side)

Equation (18) represents steady-state mass balance for the transport of gas molecules at the feed side of the natural gas mixture [30].
(18)$$D_{i,t}\left[\frac{1}{r}\frac{\partial }{\partial r}\left(r\frac{{\partial C}_{i,t}}{\partial r}\right)+\frac{\partial^{2}C_{i,t}}{\partial r}\right]=\frac{\partial }{\partial z}\left(V_{z,t}C_{i,t}\right)+\frac{1}{r}\frac{\partial }{\partial r}\left({rv}_{r,t}C_{i,t}\right),$$
where *i* represents every component of the gas mixture. The following equation describes the velocity profile in the tube.

For laminar flow inside the tube [35]:(19)$$V_{Z,t}=v_{Z,t,max}\left(1-{\left(\frac{r}{R}\right)}^{2}\right),$$

The boundary conditions [33]:

Inlet gas conditions: *Z* = 0
(20)$$C_{{CO}_{2},t}=C_{{CO}_{2}}^{o},\;\;\;\;C_{{CH}_{4,t}}=C_{{CH}_{4}}^{o},$$

Outlet gas conditions: Z=L
(21)$$nN=0,\;\;\;\;N_{i}=-D_{i,t}\nabla C_{i,t}+C_{i,t}u_{i,t},$$

At *r* = 0
(22)$$-\frac{{\partial C}_{i,t}}{\partial r}=0,$$

At *r* = *r*_1_
(23)$$C_{i,t}=C_{i,m}/K,$$
(24)$$N_{i}=-D_{i,t}\nabla C_{i,t}+C_{i,t}u_{r,t}$$

#### 3.1.2. Membrane Part

The steady-state mass balance for the transport of gas molecules from the gas mixture across the membrane skin layer is considered to be due to diffusion. Hence, the derived equation is [32]:(25)$$D_{i,m}\left[\frac{\partial^{2}C_{i,m}}{{\partial r}^{2}}+\frac{1}{r}\frac{{\partial C}_{i,m}}{\partial r}+\frac{\partial^{2}C_{i,m}}{{\partial z}^{2}}\right]=0,$$

The boundary conditions:

Membrane interface (considering solubility):(26)$$r=r_{1},\;\;\;\;C_{i,m}={C_{D}}_{i}+C_{Hi}=S_{i}p_{i}+{\overset{´}{C_{H}}}_{i}b_{i}p_{i}/\left(1+b_{i}p_{i}\right),$$

At *r* = *r*_1_
(27)$$C_{i,m}=C_{i,t}\times K,$$
(28)$$N_{i}=-D_{i,m}\nabla C_{i,m},$$

At *r* = *r*_2_
(29)$$C_{i,m}=C_{i,s}\times K,$$
(30)$$N_{i}=-D_{i,m}\nabla C_{i,m},$$

At *z* = 0, *z* = *L*
(31)n·N=0,
(32)$$N_{i}=-D_{i,m}\nabla C_{i,m}.$$

#### 3.1.3. Permeate Side

The steady-state material balance for the transport of gas molecules from the gas mixture on the permeate side is considered to be due to diffusion and convection [32].
(33)$$D_{i,p}\left[\frac{1}{r}\frac{\partial }{\partial r}\left(r\frac{{\partial C}_{i,p}}{\partial r}\right)+\frac{\partial^{2}C_{i,p}}{\partial r}\right]=\frac{\partial }{\partial z}\left(V_{z,p}C_{i,p}\right)+\frac{1}{r}\frac{\partial }{\partial r}\left({rv}_{r,p}C_{i,p}\right),$$

Equation (34) was used to minimize the deviation into the velocity profile on the shell side [34]:(34)$$V_{z,p}=v_{z,pmax}\left[\frac{{\left(\frac{r}{r_{4}}\right)}^{2}-{\left(\frac{r_{3}}{r_{4}}\right)}^{2}+2ln\left(\frac{r_{3}}{r}\right)}{3+{\left(\frac{r_{3}}{r_{4}}\right)}^{4}-4{\left(\frac{r_{3}}{r_{4}}\right)}^{2}+4ln\left(\frac{r_{3}}{r_{4}}\right)}\right],$$

At *r* = *r*_3_
(35)$$C_{i,p}=C_{i,s},$$
(36)$$N_{i}=-D_{i,p}\nabla C_{i,p}+C_{i,p}u_{i,p},$$

At *z* = 0, *z* = *L*
(37)n·N=0,
(38)$$N_{i}=-D_{i,p}\nabla C_{i,p}+C_{i,p}u_{i,p}.$$

Equations (39)–(50) represent the Navier–Stokes equation with boundary conditions which were used to estimate the velocity profiles at tube and shell sides:

#### 3.1.4. Feed Side (Tube Side)

*r*-direction:(39)$$\rho_{g}\left(v_{r,t}\frac{{\partial v}_{r,t}}{\partial r}+v_{z,t}\frac{{\partial v}_{z,t}}{\partial z}\right)=-\frac{{\partial p}_{t}}{\partial r}+\mu_{g}\left[\frac{\partial }{\partial r}\left(\frac{1}{r}\frac{\partial }{\partial r}\left({rv}_{r,t}\right)+\frac{\partial^{2}v_{r,t}}{{\partial z}^{2}}\right)\right],$$

*z*-direction:(40)$$\rho_{g}\left(v_{r,t}\frac{{\partial v}_{z,t}}{\partial r}+v_{z,t}\frac{{\partial v}_{z,t}}{\partial z}\right)=-\frac{{\partial p}_{t}}{\partial z}+\mu_{g}\left[\frac{\partial }{\partial r}\left(\frac{1}{r}\frac{\partial }{\partial r}\left({rv}_{,t}\right)+\frac{\partial^{2}v_{z,t}}{{\partial z}^{2}}\right)\right]+g,$$

At *r* = 0
(41)$$-\frac{{\partial v}_{z,t}}{\partial r}=0,\;\;\;v_{r,t}=0,$$

Velocity near membrane walls:(42)$$r=r_{1},\;\;v_{z,t}=0,\;\;v_{r,t}=0,$$

Inlet velocity of feed gas:(43)$$z=0,\;\;v_{z,t}=v_{o,t},\;\;v_{r,t}=0,$$

Outlet velocity of feed gas:(44)$$z=L,\;\;\;\;\frac{-{\partial v}_{z,t}}{\partial z}=0,\;\;\;\;v_{r,t}=0.$$

#### 3.1.5. Shell Side

The Navier–Stockes equation is given:

*r*-direction:(45)$$\rho_{g}\left(v_{r,s}\frac{{\partial v}_{r,s}}{\partial r}+v_{z,s}\frac{{\partial v}_{z,s}}{\partial z}\right)=-\frac{{\partial p}_{s}}{\partial r}+\mu_{g}\left[\frac{\partial }{\partial r}\left(\frac{1}{r}\frac{\partial }{\partial r}\left({rv}_{r,s}\right)+\frac{\partial^{2}v_{r,s}}{{\partial z}^{2}}\right)\right],$$

*Z*-direction:(46)$$\rho_{g}\left(v_{r,t}\frac{{\partial v}_{z,s}}{\partial r}+v_{z,s}\frac{{\partial v}_{z,s}}{\partial z}\right)=-\frac{{\partial p}_{s}}{\partial z}+\mu_{g}\left[\frac{\partial }{\partial r}\left(\frac{1}{r}\frac{\partial }{\partial r}\left({rv}_{z,s}\right)+\frac{\partial^{2}v_{z,s}}{{\partial z}^{2}}\right)\right]+g,$$

Boundary conditions utilized:

Membrane gas interface:(47)$$r=r_{3},\;\;v_{z,s}=0,\;\;v_{r,s}=0,$$

Axial symmetry at module hypothetical radius:(48)$$r=r_{3},\;\;-\frac{{\partial v}_{z,s}}{\partial z}=0,\;\;v_{r,s}=0,$$

The outlet of the gas flow in the shell side:(49)$$z=L,\;\;-\frac{{\partial v}_{z,s}}{\partial z}=0,\;\;v_{r,s}=0,$$

Inlet gas velocity:(50)$$z=0,\;\;v_{z,s}=0,\;\;v_{r,s}=0.$$

Operation conditions, membrane characteristics, diffusion coefficients, partition factors, and the other transport properties that were used for solving the mass transfer equations are given in Table 3.

### 3.2. Numerical Procedure

The model equations regarding the shell side, membrane, and tube side with the proper boundary conditions were solved utilizing COMSOL Multiphysics (5.6) software, using the technique of finite element (FEM). The time for treating the collection of equations was around 43 s. Figure 4 illustrates a part of the mesh utilized to decide the gas transport behavior in a hollow fiber membrane (HFM). It must be indicated that the COMSOL mesh generator makes triangular meshes that are isotropic in volume. Great numbers of components are then made with scaling. A scaling parameter of 300 was used in the *z*-direction due to a large variance between r and z. COMSOL spontaneously tabulates back the geometry after meshing. This makes an additional fine mesh of about 76,742 degrees of freedom solved and 2384 internal DOFs.

## 4. Experimental Work

### 4.1. Materials and Experimental Design

In this study, a commercial membrane-type MCB-1512A purchased from the Korean company Airrane was used. The membrane, consisting of polysulfone, possesses high selectivity in carbon dioxide separation but low permeance. The low permeability problem is overcome by making the membrane fibers asymmetrically composed of several layers. All information related to this model of membranes has been provided by the company in Table 4.

In addition to the information provided by the manufacturer, a Scanning Electron Microscopy (SEM) test was performed for the fibers, wherein the number of layers that make up each fiber, as well as the dimensions of each layer was determined. The SEM images of the hollow fibers showed an external diameter of 351.51 µm and an internal diameter of 219.87 µm (Figure 5).

Through the cross-section of the fiber shown in Figure 5, the thickness of the dense layer is 20 μm and the thickness of the support layer is 40 μm.

In this work, three independent variables were selected: pressure feed, temperature feed, and the percentage of carbon dioxide in the feed. Meanwhile, the concentration of carbon dioxide in the permeate and flux are the two dependent variables. The selection of the parameters that were studied was based on the design limits of the membrane used for both pressure and temperature. As for the percentage of carbon dioxide, it was based on the analysis of natural gas in the fields of the Maysan Oil Company. The limits and steps of all these variables are based on which of the experiments were designed in the MINITAB program and according to the type of Taguchi, which are shown in the following Table 5.

In all the experiments, the feed flow rate was adjusted at 3.5084 L/min. At steady-state conditions, the permeate stream was sent to (Gas chromatography) GC, and its mole fraction (volume fraction) was measured, while the permeate flow was measured using a bubble meter. Table 6 represents the number of runs with the results obtained after conducting the experiments, and it includes the percentage of carbon dioxide in the reject.

### 4.2. Lab Scale System and Gas Analyzers

Experiments were conducted to separate carbon dioxide from the gas mixture using a laboratory system, shown in Figure 6.

In this study and during laboratory experiments, the gas is analyzed by taking samples from two points in the laboratory system; the first is from the gas mixture feeding the membrane, and the second is from the gas mixture separated by the membrane. In the first point, the gas sample is analyzed by the Dräger Short-term Tubes. The second sample is analyzed by connecting the permeate side with the GC device using an online connection.

## 5. Results and Discussion

In this work, the flux was calculated once from the mathematical model and again from the results of the simulation model that was developed by COMSOL 5.6 and compared between them. Calculating the flux utilizing the resistance in a series model, represented by Equation (4), requires finding the diffusivity through the different domains of the membrane, as well as the mass transfer coefficient, permeability, and solubility. The effect of operating conditions in this study on the mentioned parameters was investigated as follows.

### 5.1. Effect of Pressure and Temperature on Diffusion Coefficients

The diffusivity of each component was calculated using Hirschfelder’s equation for the binary mixture [37]. In this work, the impacts of temperature and pressure on the coefficients of diffusion were investigated in both the feed and permeate regions. Figure 7 and Figure 8 show the influence of pressure and temperature on the diffusivity in the feed side of the two gases separately in the mixture.

Figure 7 plots the influence of temperature on the coefficient of diffusion of gases in the feed side at the temperature range of 293–318 K. The coefficient of diffusion of gases rises with enhancing temperature. This behavior can be justified by Hirschfelder’s equation [37]. Regarding the effect of pressure, generally, the coefficient of diffusion is inversely proportional to the pressure, and this is proved by Figure 8 [38,39].

On the permeate side, the pressure value is constant at one atmosphere for all feed pressure values, while the temperature is similar to the temperature of the gas mixture entering the separation unit. The change in diffusion here corresponds to the change in the feeding side under the same temperature with constant pressure at atmospheric pressure.

On the other hand, the increase in the amount of carbon dioxide in the gas mixture has a slight adverse effect on the diffusion of the gas itself and a slight direct change on the diffusion of methane. Figure 9 shows that the gas diffusivity changes with an increase in mole percent of CO_2_ in the feed at constant pressure at 5 bar and temperature at 303 K.

### 5.2. Effect of Temperature and Pressure on Mass Transfer Coefficients

The mass transfer coefficient at both the feed side and permeate side depends on the hydrodynamic of the fluid, calculated from the Reynolds, diffusivity of gas, and Sherwood and Schmidt’s numbers using Equations (11)–(13). On the feed side, the inner diameter of the fiber is used, while on the permeate side, the hydraulic diameter (Equation (16)) is used in both equations. Figure 10 shows the impact of temperature on the mass transfer coefficient of feed gases at constant pressure.

Figure 10 plots the influence of temperature on the coefficient of mass transfer of gases in the feed side at a temperature range of 293–318 K. Increasing the temperature leads to a change in the values of the parameters on which the value of the mass transfer coefficient depends. When the temperature changes from 293 K to 318 K, the value of the parameters changes in different proportions, and the Reynolds number and Sherwood number decrease by 14.1% and 7.35%, respectively.

On the other hand, the diffusivity and viscosity values increase by 16.67% and 7.1%, respectively, while the value of the Schmidt number remains constant. The behavior of the mass transfer coefficient with a change in pressure with constant temperature and gas composition is illustrated in Figure 11.

Figure 11 depicts that the coefficient of mass transfer of the two gases is related to an exponential relationship with the pressure, and its value decreases with the increase in the pressure of the feed.

Figure 12 shows the mass transfer coefficient of the gases change with the increase in mole percent of CO_2_ in the feed at constant pressure at 5 bar and temperature at 303 K. From this figure, the behavior of the coefficient of mass transfer is very similar to the behavior of diffusivity when changing the proportion of CO_2_ in the feed.

On the permeate side, the value of the coefficient of mass transfer differs compared to the feed side for two main reasons, the first is the difference in gas concentrations, as well as the stability of pressure at one atmosphere, and the second reason is the adoption of the hydraulic diameter of the outer circumference of the membrane fibers when calculating the coefficient of mass transfer.

In this domain, the value of the mass transfer coefficient is equal for both gases with the change in temperature and carbon dioxide concentration. The relationship between the mass transfer coefficient, temperature, and gas concentration is a direct relationship on this side of the membrane.

### 5.3. The Diffusion Coefficient of Gases in the Dense Membrane

The solution to mass transfer equations is based on calculating the diffusivity value of the gas mixture species in the membrane. Simply put, the voids between the polymeric chains in the membrane provide a passage path for the permeating gas molecules. The free volume theory has usually been utilized to evaluate the coefficient of diffusion of gaseous species out of different polymeric membranes [27].

In this present study, the diffusivity of carbon dioxide and methane was calculated in two stages. In the first, the Doltile relationship was used, in which the effect of temperature does not appear [23,39]. As for the second stage, the influence of temperature on the diffusivity of gases was introduced using the Arrhenius relationship. The diffusivity values are 2.28787 × 10^−8^ cm^2^/s and 3.08508 × 10^−9^ cm^2^/s for each carbon dioxide and methane, respectively, calculated from Equation (8). On the other hand, the temperature change within the range used in this study on the diffusion of gases was investigated using Equation (10). Figure 13 represents the effect of temperature on the diffusivity of the gases that consist of the feed mixture.

By analyzing the results in the above diagram, the ratio of carbon dioxide diffusion to the methane gas diffusivity decreased from 5.01 to 3.98 when the temperature changed from 293 K to 318 K, while the change in diffusivities for each degree of temperature was 1.06408 × 10^−9^ for CO_2_ and 2.93641 × 10^−10^ for CH_4_.

### 5.4. Model Validation

One of the most important validation tools for the proposed CFD model is to compare its results with experimental results [32]. The simulation results of the CO_2_ flux were compared with experimental results. There is a fine accordance between the CFD model and experimental results, with an utmost um relative mistake of 7.73%. Table 7 shows the fluxes obtained from both the CFD model and the mathematical model based on experimental data.

#### 5.4.1. Velocity Field

The velocity field, profile, contour, and 3D profile in the tube side of the HFM are seen in Figure 14, Figure 15, Figure 16 and Figure 17, in which the gas mix streams. The velocity profile in the tube side of the HFM was simulated by solving Navier–Stokes’s equations.

The profile of velocity is usually parabolic, with a maxima velocity that rises along the length of the membrane from 0.34683 m/s in the inlet to 0.45083 m/s in the outlet. Moreover, it reveals that at the entrance zones on the feed side, the velocity is undeveloped. After distancing from the entrance, the profile of velocity is totally developed. As seen, the model counts the inlet effects on the fluid flow hydrodynamics on the feed side. The velocity distribution on the shell side is obtained by solving the Navier–Stoke equations and material balance equations in conjunction with Happel’s free surface model. This mode is mainly used for HFM and proposes a parabolic velocity profile for the flow outside of the fiber (Equation (34)). The velocity field, profile, contour, and 3D profile in the shell side of the HFM are seen in Figure 18, Figure 19, Figure 20 and Figure 21.

From the velocity profile, the average velocity rises with the membrane length due to the continuous CO_2_ gas permeation. The average velocity for any step of the *Z* value from the total length of the fiber can be calculated from the following formula:(51)$$V_{z-shell}=\frac{\int_{r2}^{r3}{V(r)}_{z-shell}dA}{\int_{r2}^{r3}dA}.$$

Calculating the average velocity at both ends of the fiber, its value was 1.362 × 10^−4^ (m/s) at the closed end of the shell, while the average velocity at the exit was 0.084923 (m/s). The increase in this speed is due to the flux of carbon dioxide gas through the membrane from the feed side to the shell side during the separation process.

#### 5.4.2. The Concentration Distribution of Gas in the Membrane

The equations of continuity, moment, and mass transfer for the three domains in the membrane model were solved for both carbon dioxide and methane. Figure 22 describes the CO_2_ concentration gradient in the tube, the membrane, and the shell of the hollow fiber membrane.

When the gas moves along the tube, CO_2_ transmits to the membrane due to the concentration gradient. Considering Figure 16, at *z* = 0 where the gas inflows the HFM, the CO_2_ concentration has a peak value amounting to 11.909 $mol/m^{3}$, while on the permeate side of the membrane, the average concentration of CO_2_ is equal to 11.723 $mol/m^{3}$ at *z* = *L*. The mechanisms of mass transfer in the tube and the shell are diffusion and convection. Since the flow is in the *z*-direction, the mass is transmitted by convection. In the radial direction, diffusion performs the main role in the phenomena of mass transfer. Carbon dioxide gas flows by the diffusion mechanism, where it is absorbed on the surface of the membrane and transferred to the other side [32]. Figure 23 illustrates a 2D concentration difference with the overall flow vectors of CO_2_. Moreover, a 3D concentration gradient of CO_2_ is seen in Figure 24, only for the best conception of the transfer of mass.

It is also possible through the COMSOL model to obtain accurate results about the mass transfer, especially in the area of the boundary layer and the concentrations of carbon dioxide on the two faces of the membrane. Figure 25 shows a one-dimensional concentration gradient of carbon dioxide through the region of the tube, membrane, and shell at 15% of the fiber length. At 0L (fiber entrance) of the tube length, the mass transfer of the CO_2_ was only on the first surface of the membrane and at a concentration of 8.8605 $mol/m^{3}$. After 5% of the tube length, there was a noticeable transfer of carbon dioxide gas to the permeate side of the membrane, where the gas concentrations on the first and second surfaces of the membrane were 9.35 and 7.1083 $mol/m^{3}$, respectively. Table 8 presents the concentrations of carbon dioxide on the sorption side and desorption side in the membrane.

By making simple calculations for the results in Table 8, it can be noted that the carbon dioxide concentration has risen to 95% of the highest concentration (9.539 $mol/m^{3}$) of the gas on the desorption surface. This indicates that the effectiveness of mass transfer to length is very high at the beginning of the fiber.

### 5.5. Analysis of CO_2_ Flux and Recovery

The experimental results in Table 7 were analyzed by Minitab.18 to discover the effect of pressure, temperature, and concentration on the flux of carbon dioxide through the membrane. Figure 26 illustrates the effect of pressure on the flux of CO_2_ in the membrane at a pressure range of 2–10 bar. It is clear in Figure 26 that the flux increases with increasing pressure, due to increasing the equilibrium concentration of carbon dioxide on the surface of the membrane. The increase in the concentration difference on the two surfaces of the membrane represents an increase in the driving force in the law of flux.

Figure 27 shows the effect of carbon dioxide concentration in the feed on the flux through the membrane. The relationship of flux to concentration is a direct type, as shown in the Figure. The reason for this is that the mass transfer across the two faces of the membrane is due to the concentration difference, and the higher the numerical value of this difference the higher the transfer rate.

The effect of the third factor represented by temperature is almost non-existent on flux, due to the effect of temperature inversely on the gas solubility of the membrane polymer and directly on the diffusion of the gas across the membrane. Since permeability is the product of these two factors, the temperature does not affect flux.

After analyzing the results, the role of the most important step comes, which is to find the optimal conditions that guarantee the highest recovery in the membrane. The optimum conditions are found to be 7.5 bar for pressure, 293 K for temperature, and 10% for CO_2_ concentration, while at a pressure of 2.5 bar, 293.57 K, and 2% CO_2_, flux and recovery would be at non-optimal conditions.

## 6. Conclusions

In this work, the capture of carbon dioxide using a dense hollow fiber membrane was studied experimentally and theoretically. Twenty-seven experiments have been conducted using a mixture of methane and carbon dioxide to simulate natural gas. A comprehensive mathematical model was developed to describe the flux of CO_2_ through the membrane. The flux was calculated, taking into consideration the effect of pressure, temperature, and concentration on the properties of the gas mixture, such as density, viscosity, diffusivity, solubility, and mass transfer coefficient. Through the results, the two most controlling factors in the mass transfer were pressure and concentration. Then, a 2D axisymmetric model of a multilayer hollow fiber composite membrane for CO_2_ segregation was suggested. The model considers the axial and radial diffusion in the HFM. CFD mechanisms were adopted to work out the model equations including continuity and momentum equations. The CFD model predicted the two-dimensional velocity, pressure, and concentration profiles in three domains of the fiber. Modeling forecastings were supported by the experimental results, and a reasonable harmony between them was observed. The relative error between the results of the mathematical model and the CFD model in calculating the flux ranged from 0.391 to 7.73%. The results also showed the direct effect of each of the pressures and the concentration of carbon dioxide in the feed on the flux, while the feed temperature had no obvious effect. The developed model can be used to predict the performance of membranes made of different polymers as well as other operational conditions. The limitation of this upgraded model is its use of low carbon dioxide concentrations as well as pressures that do not exceed 15 bar for the feed.

## Figures and Tables

**Figure 1 membranes-13-00557-f001:**
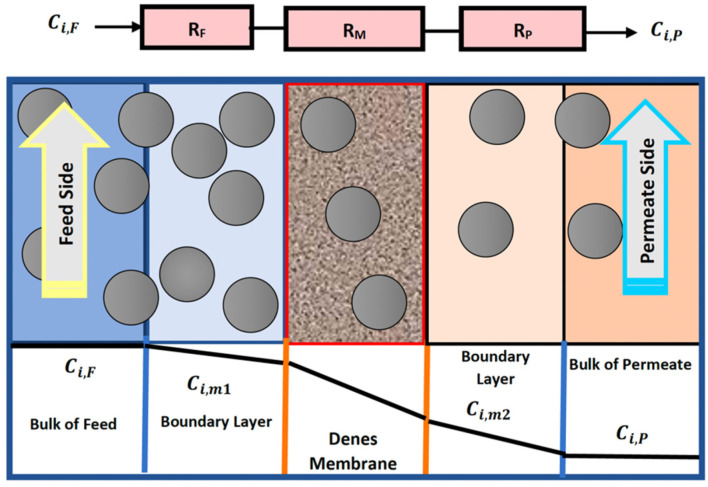
Concentration and partial profiles of CO_2_ transport through the nonporous membrane.

**Figure 2 membranes-13-00557-f002:**
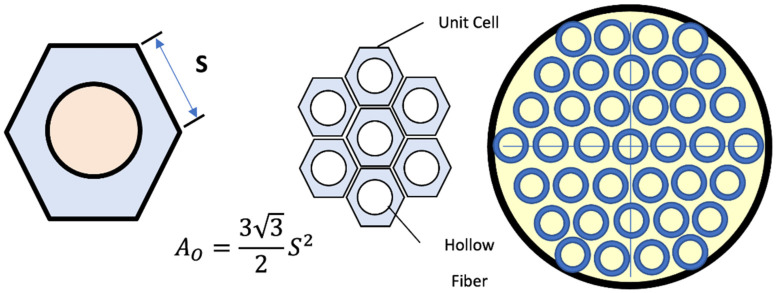
A schematic diagram for the hollow fiber membrane & cross-sectional area of permeate side [32].

**Figure 3 membranes-13-00557-f003:**
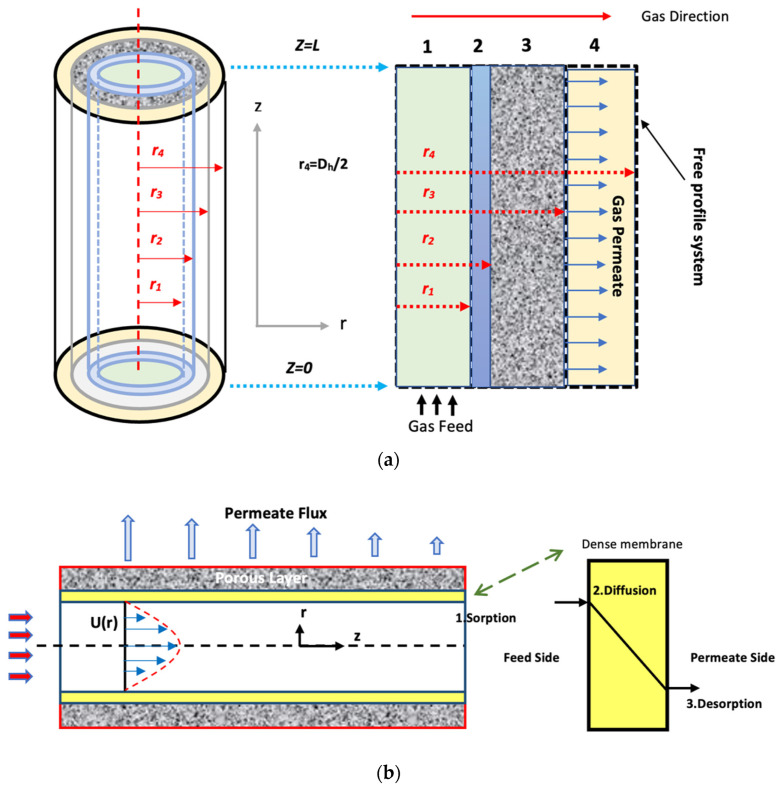
(**a**) Schematic of the fiber [33] and (**b**) The flow geometry and the steps of the solution diffusion theory [34].

**Figure 4 membranes-13-00557-f004:**
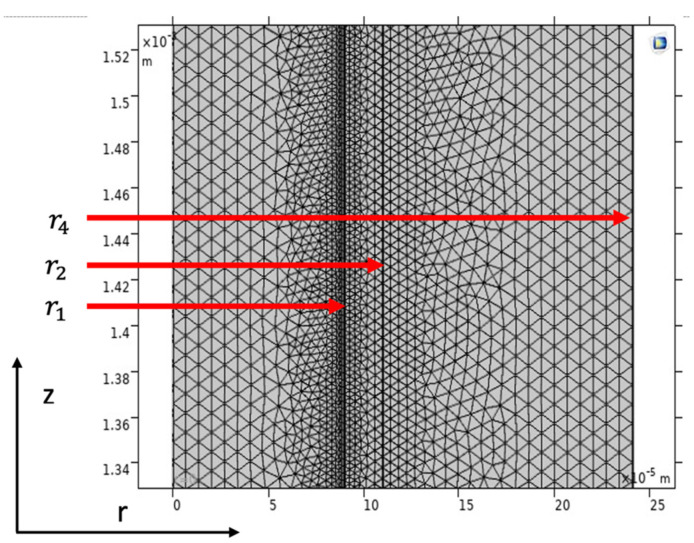
Model domain and meshes used for simulation.

**Figure 5 membranes-13-00557-f005:**
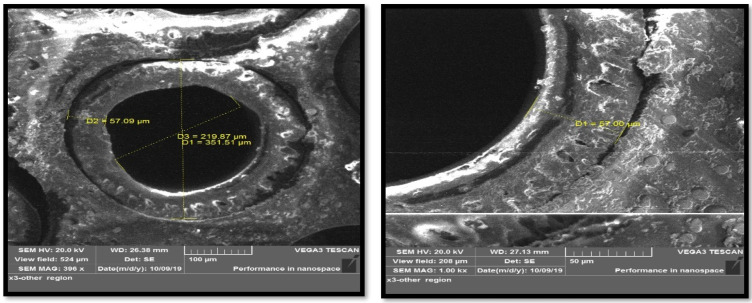
SEM snapshots of hollow fiber membrane.

**Figure 6 membranes-13-00557-f006:**
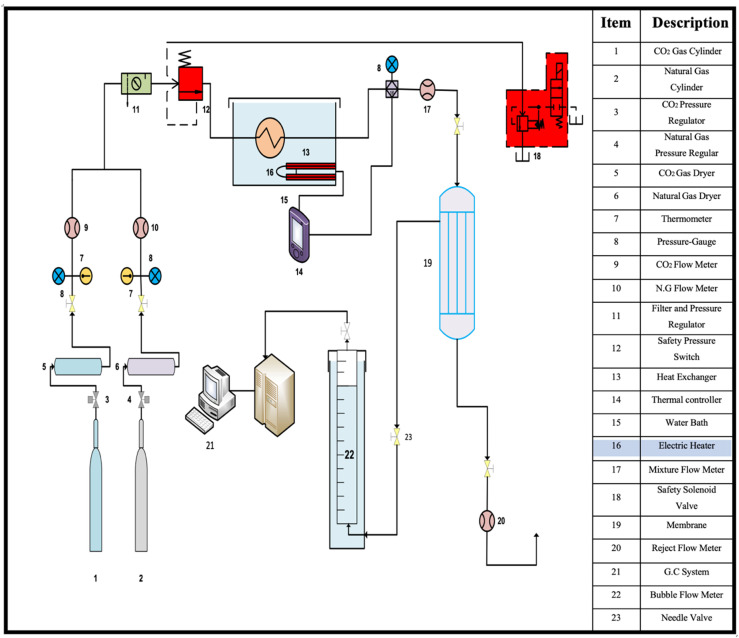
Flow diagram of the lab-scale system.

**Figure 7 membranes-13-00557-f007:**
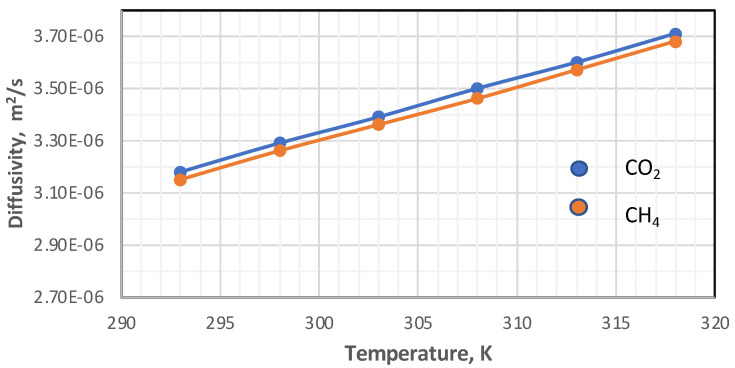
Influence of the gas temperature on coefficients of diffusion in feed side at *P_f_* = 5 bar, CO_2_ = 6 mol%, and CH_4_ = 94 mol%.

**Figure 8 membranes-13-00557-f008:**
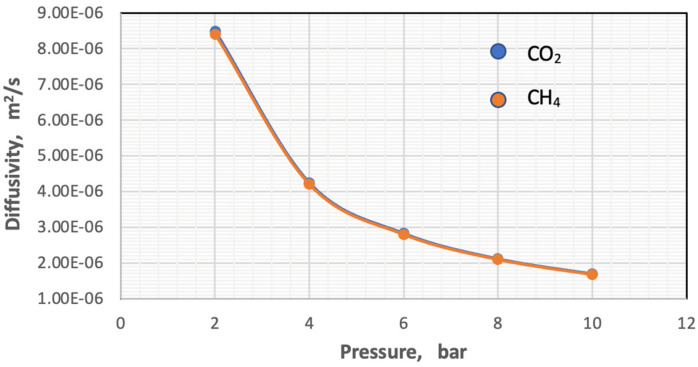
Influence of the pressure on coefficients of gases in feed side at *T_f_* = 303 K, CO_2_ = 6 mol%, and CH_4_ = 94 mol%.

**Figure 9 membranes-13-00557-f009:**
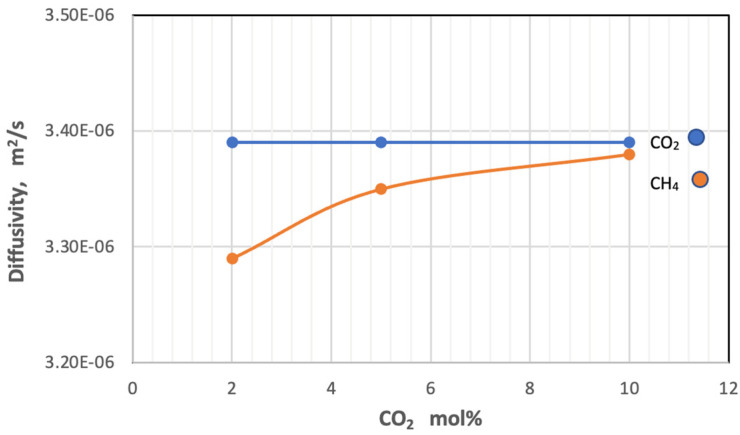
Effect of the percentage of CO_2_ on coefficients of diffusion of gases in feed side at *P_f_* = 5 bar and *T_f_* = 303 K.

**Figure 10 membranes-13-00557-f010:**
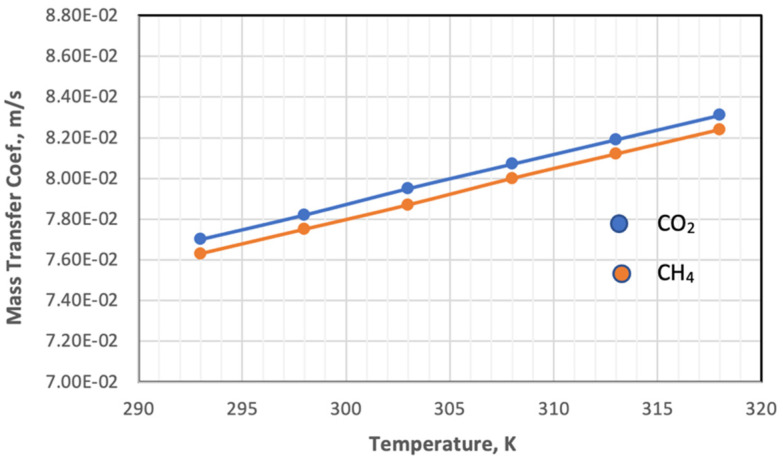
Influence of the temperature on the coefficient of mass transfer of gases in feed side at *P_f_* = 5 bar, CO_2_ = 6 mol%, and CH_4_ = 94 mol%.

**Figure 11 membranes-13-00557-f011:**
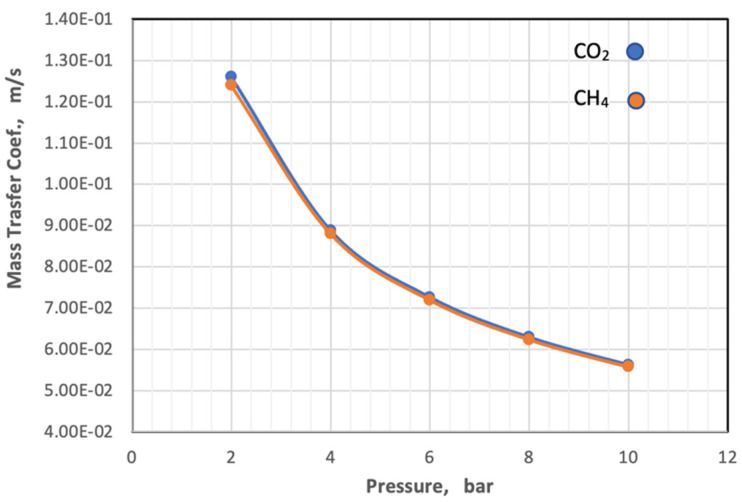
Influence of the pressure on coefficient of mass transfer of gases in feed side at *T_f_* = 303 K, CO_2_ = 6 mol%, and CH_4_ = 94 mol%.

**Figure 12 membranes-13-00557-f012:**
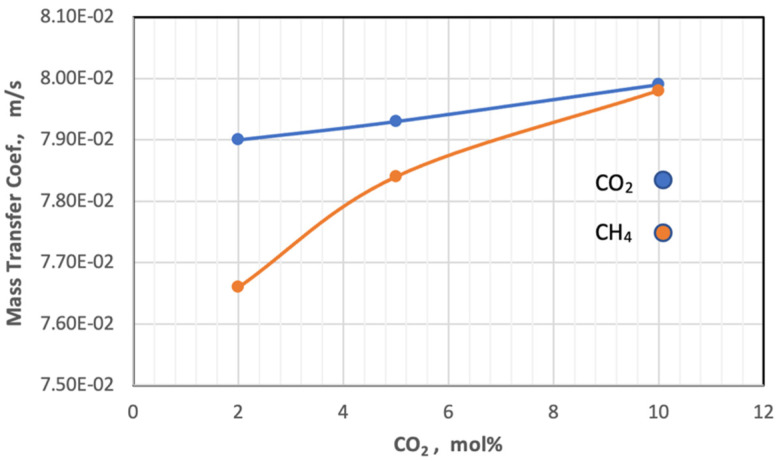
Effect of the percentage of CO_2_ on mass transfer coefficient of gases in fees side at *P_f_* = 5 bar and *T_f_* = 303 K.

**Figure 13 membranes-13-00557-f013:**
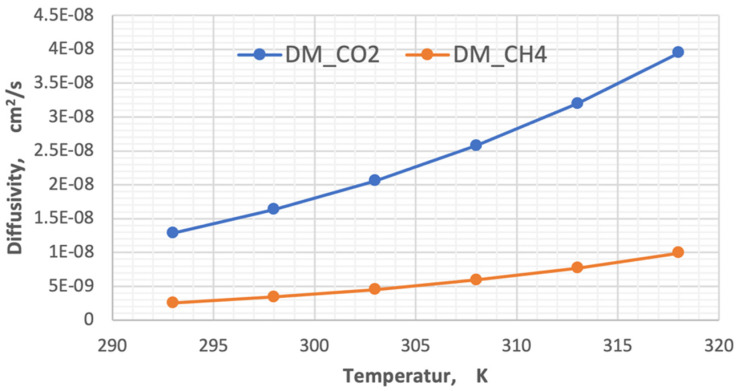
Influence of the temperature on the diffusivity of CO_2_ and CH_4_.

**Figure 14 membranes-13-00557-f014:**
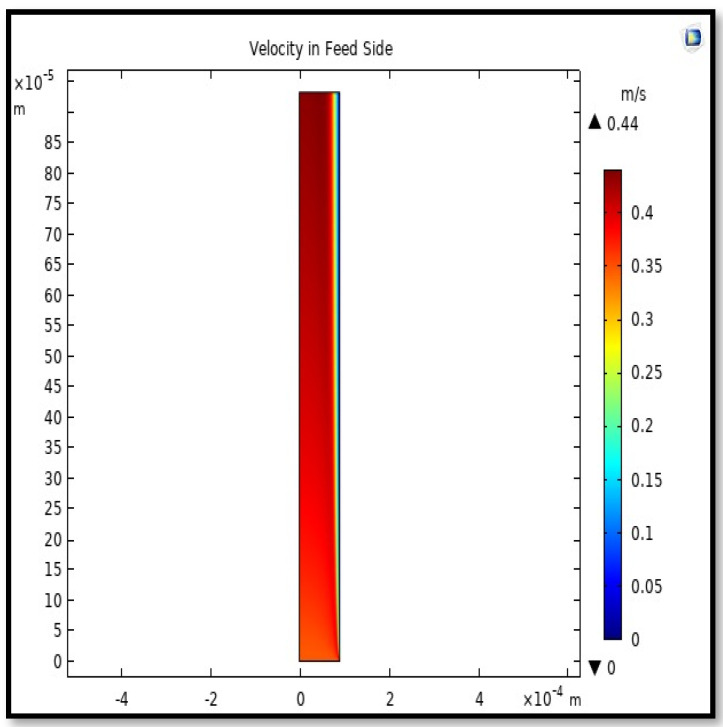
Field of velocity in the feed-side of the HFMC. Flow of gas = 3.333 × 10^−5^ (m^3^/s), *T_f_* = 303 K, and *P_f_* = 5 bar.

**Figure 15 membranes-13-00557-f015:**
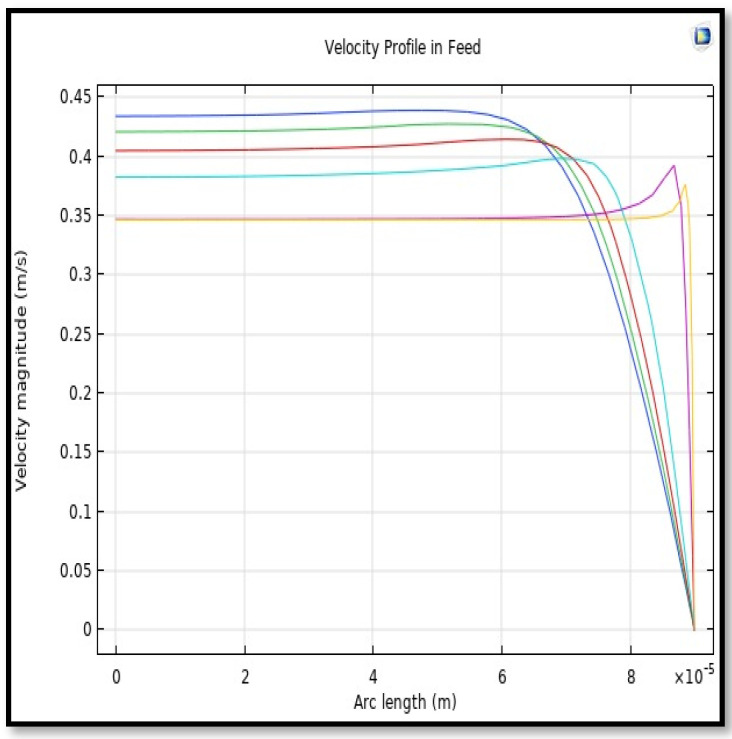
Profile of velocity in the feed-side across the length of membrane; Flow of gas = 3.333 × 10^−5^ (m^3^/s), *T_f_* =303 K, and *P_f_* = 5 bar.

**Figure 16 membranes-13-00557-f016:**
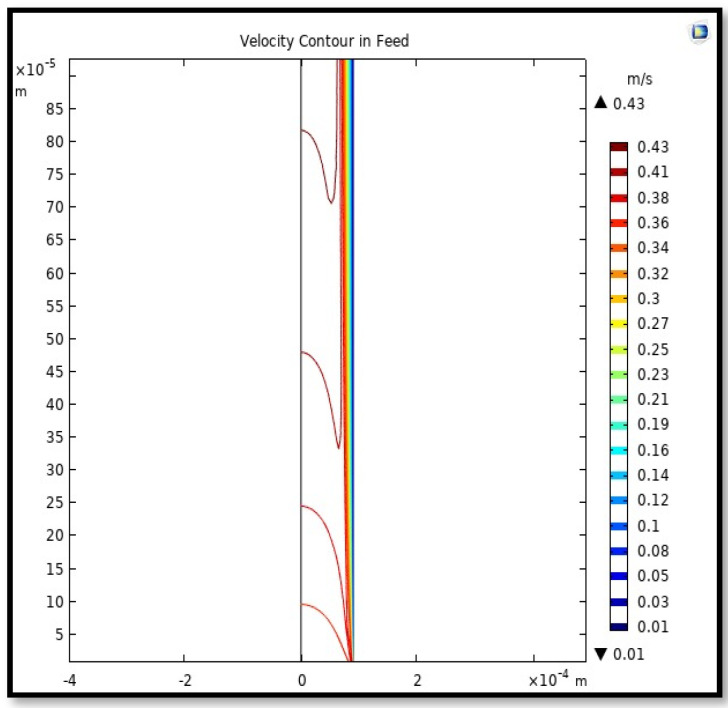
The contour of velocity in the feed-side of the HFM. Flow of gas = 3.333 × 10^−5^ (m^3^/s), *T_f_* =303 K, and *P_f_* = 5 bar.

**Figure 17 membranes-13-00557-f017:**
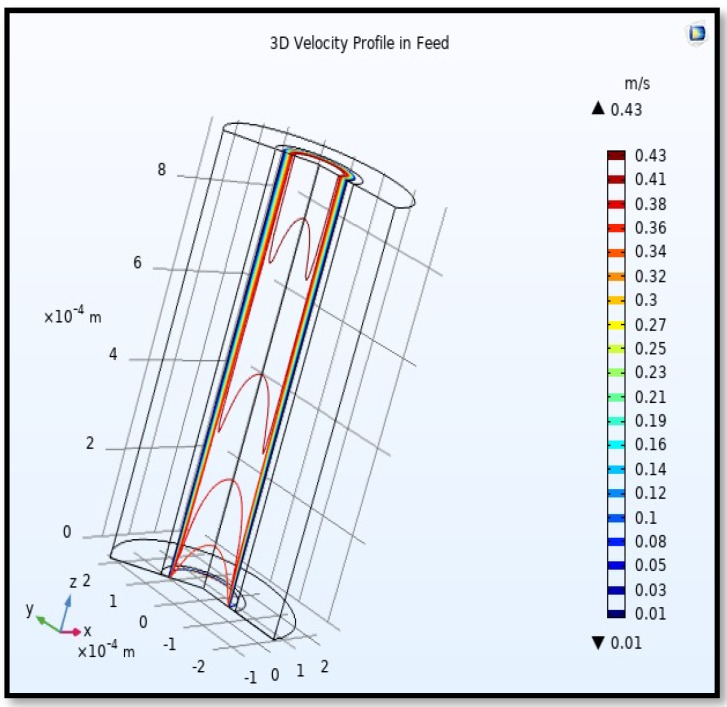
3D Profile of velocity in the feed-side across the membrane length. Flow of gas = 3.333 × 10^−5^ (m^3^/s), *T_f_* =303 K, and *P_f_* = 5 bar.

**Figure 18 membranes-13-00557-f018:**
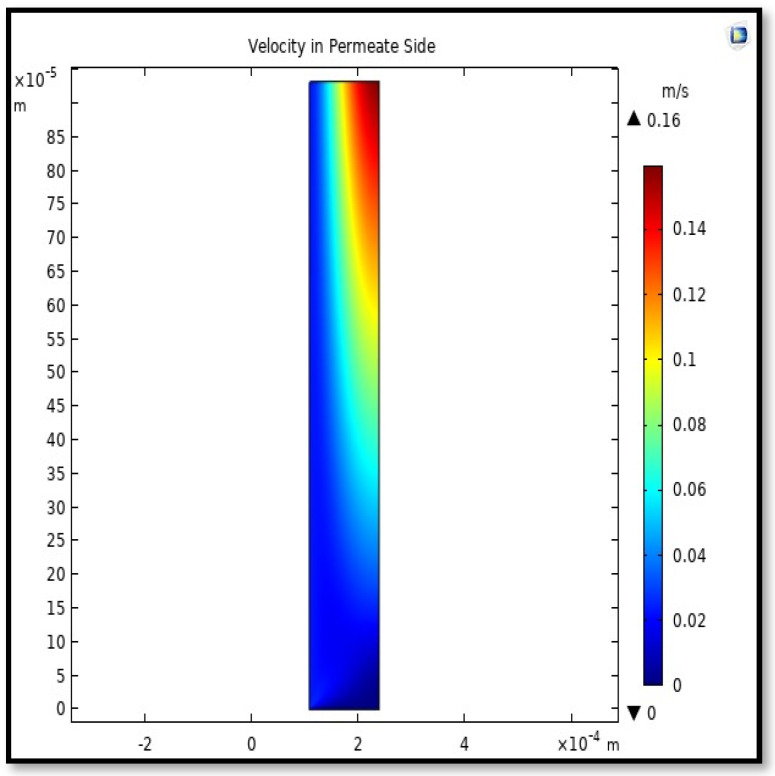
Field of velocity in the shell side of the HFMC. Flow of gas = 3.333 × 10^−5^ (m^3^/s), *T_f_* =303 K, and *P_f_* = 5 bar.

**Figure 19 membranes-13-00557-f019:**
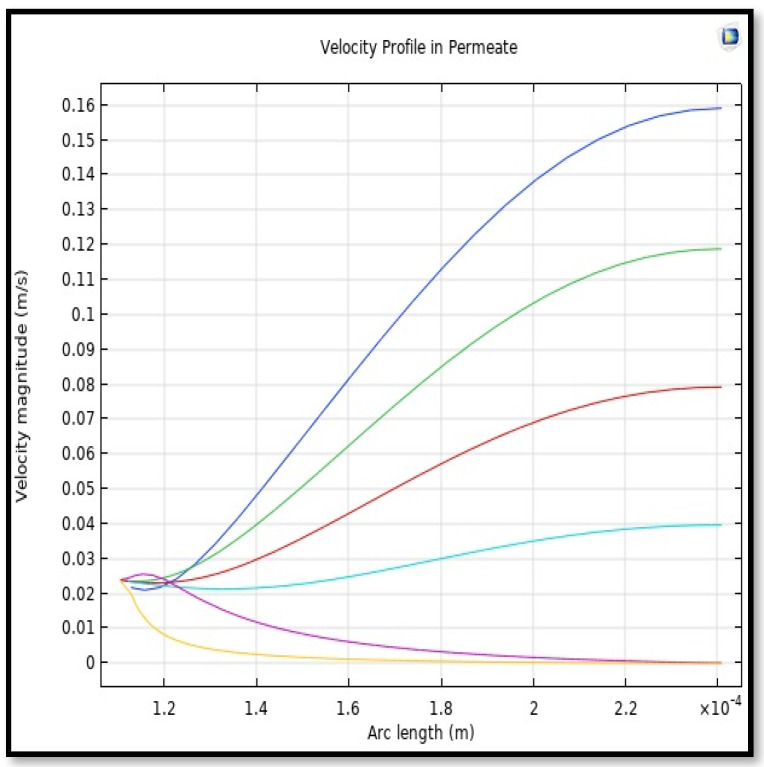
Profile of velocity in the shell side across the membrane length. Flow of gas = 3.333 × 10^−5^ (m^3^/s), *T_f_* =303 K, and *P_f_* = 5 bar. Each color represents a different section from length along the fiber.

**Figure 20 membranes-13-00557-f020:**
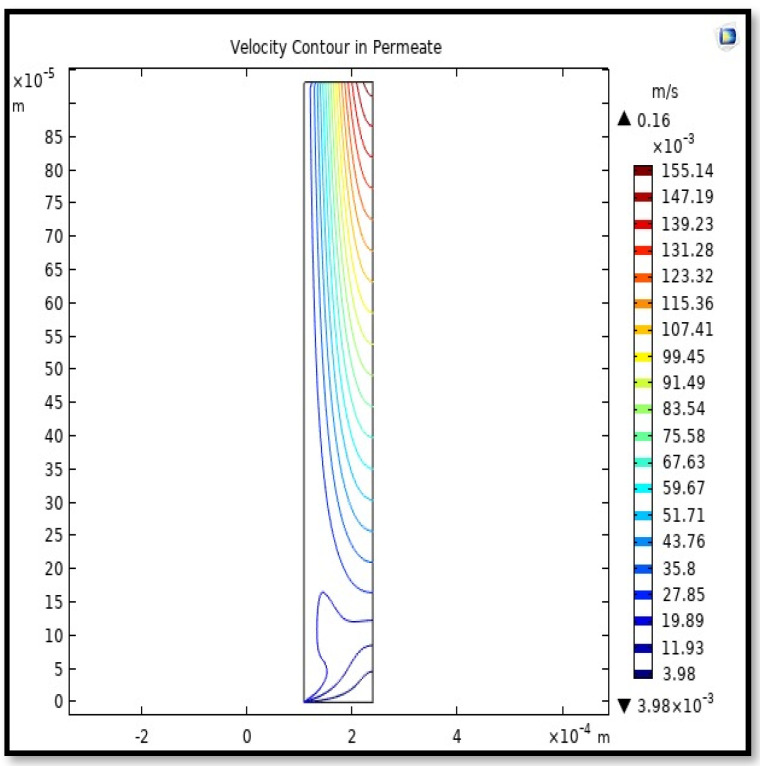
Velocity Contour in the sell side of the HFM. Flow of gas = 3.333 × 10^−5^ (m^3^/s), *T_f_* =303 K, and *P_f_* = 5 bar.

**Figure 21 membranes-13-00557-f021:**
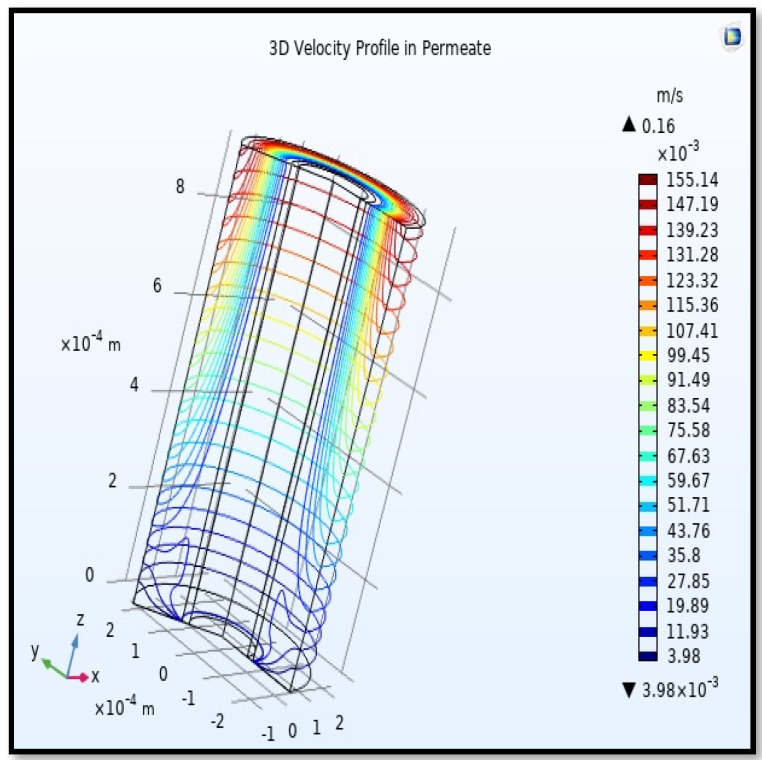
3D profile of velocity in the shell side across the membrane length. Gas flow rate = 3.333 × 10^−5^ (m^3^/s), *T_f_* =303 K, and *P_f_* = 5 bar.

**Figure 22 membranes-13-00557-f022:**
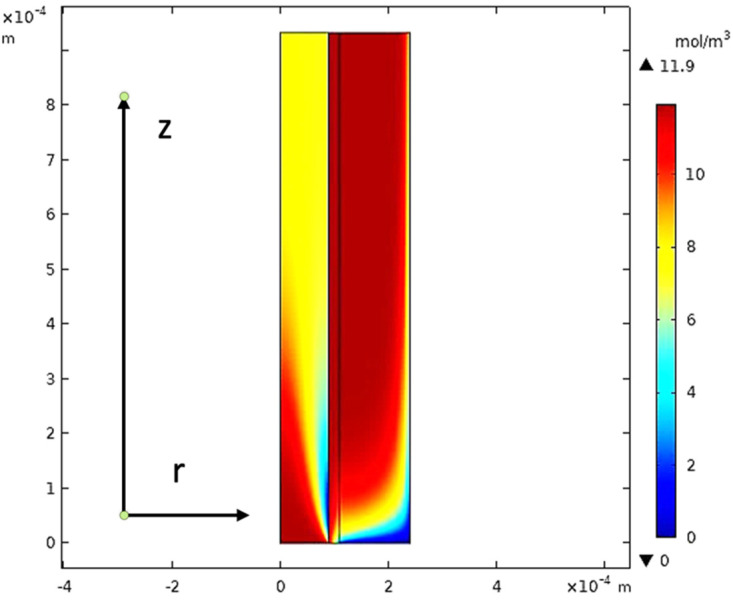
The concentration gradient of CO_2_ in the model domains at feed flow = 3.333 × 10^−5^ (m^3^/s), *T_f_* = 30 °C, *P_f_* = 5 bar, and X_CO_2__ = 0.06.

**Figure 23 membranes-13-00557-f023:**
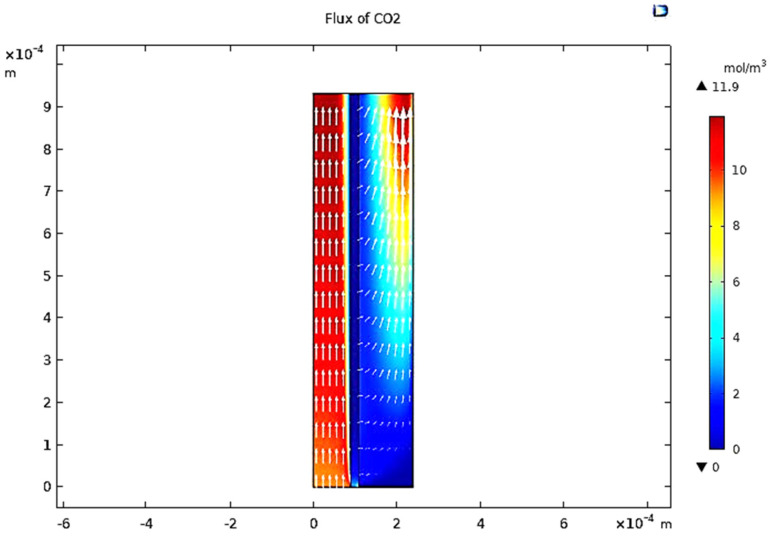
Vectors of overall flux in the sections of the model at feed flow = 3.333 × 10^−5^ (m^3^/s), *T_f_* = 30 °C, *P_f_* = 5 bar, and X_CO_2__ = 0.06.

**Figure 24 membranes-13-00557-f024:**
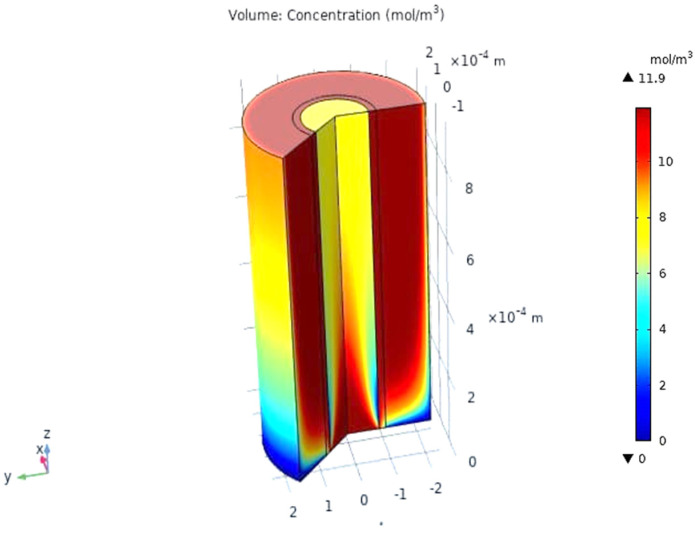
The 3D concentration gradient of CO_2_ in all domains at feed flow = 3.333 × 10^−5^ (m^3^/s), *T_f_* = 30 °C, *P_f_* = 5 bar, and X_CO_2__ =0.06.

**Figure 25 membranes-13-00557-f025:**
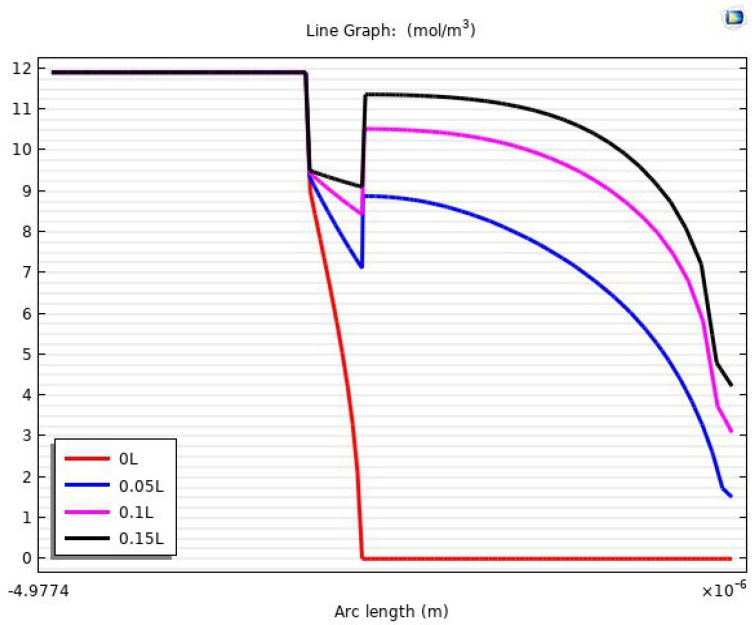
The one-dimension concentration gradient of CO_2_ in all domains.

**Figure 26 membranes-13-00557-f026:**
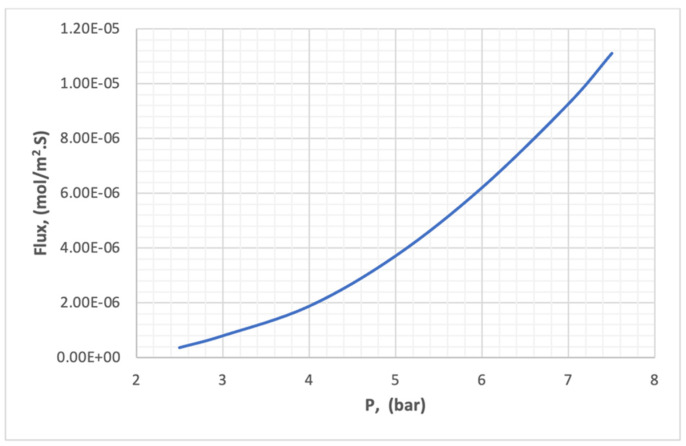
The effect of pressure on the flux of CO_2_ in the membrane.

**Figure 27 membranes-13-00557-f027:**
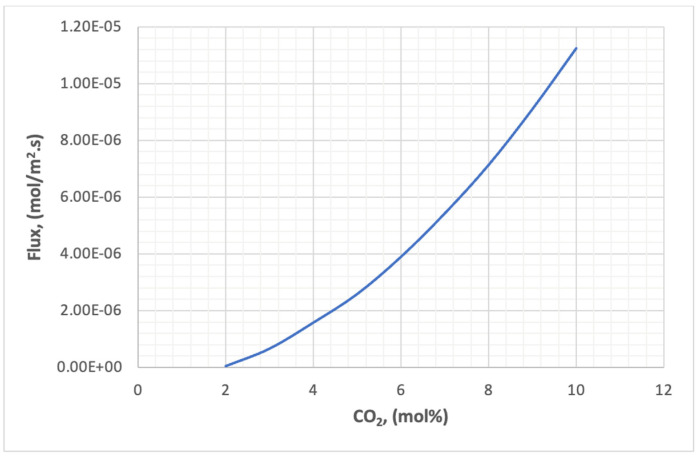
The effect of concentration of CO_2_ on the flux in the membrane.

**Table 1 membranes-13-00557-t001:** The free volume constants for the Doolittle Relation [27,28].

Gas	*A* (cm^2^ s^−1^)	*B*
CO_2_	$2.08\times 10^{-5}$	1.09
CH_4_	$5.24\times 10^{-6}$	1.19

**Table 2 membranes-13-00557-t002:** The pre-exponential factor and apparent activation energy of gas components [28].

Gas	*A* (cm^2^ s^−1^)	$E_{d}$ (Kcal/Mol)
CO_2_	0.02	8.3
CH_4_	0.074	10

**Table 3 membranes-13-00557-t003:** The input variables for the proposed model in COMSOL 5.6.

Parameter	Value	Unit	Parameter	Value	Unit
Temperature of a gas mixture	303	K	%CO_2_ in Feed	6	mol%
Pressure of a gas mixture	5	bar	%CH_4_ in Feed	94	mol%
Feed Flow Rate	3.33 × 10^−5^	$m^{3}/s$	The density of feed gas	3.59	$kg/m^{3}$
Inlet Conc. of CO_2_	1.20 × 10^1^	$mol/m^{3}$	The viscosity of the feed gas	1.15 × 10^−5^	g/cm.s
Inlet Conc. of CH_4_	1.83 × 10^2^	$mol/m^{3}$	The inner radius of the fibre	90	μm
Inlet gas velocity in fibre	0.345	m/s	The outer radius of the fibre	150	μm
Number of fibres	3800	-	The thickness of the dense layer	20	μm
Scale	300	-	Length of fibre	28	cm
Diffusion Coef. of CO_2_ in Tube Side	3.39 × 10^−5^	$m^{2}/s$	Partition Factor of CO_2_	0.801	-
Diffusion Coef. of CH_4_ in Tube Side	3.36 × 10^−6^	$m^{2}/s$	Partition Factor of CH_4_	0.441	-
Diffusion of CO_2_ in Membrane	2.29 × 10^−8^	$m^{2}/s$	Density of permeate gas	0.87	$kg/m^{3}$
Diffusion of CH_4_ in Membrane	3.1 × 10^−9^	$m^{2}/s$	Viscosity of the permeate gas	1.24 × 10^−5^	g/cm.s
Diffusion Coefficients of CO_2_ in Permeate Side	1.72 × 10^−5^	$m^{2}/s$	Diffusion Coefficients of CH_4_ in Permeate Side	1.72 × 10^−5^	$m^{2}/s$

**Table 4 membranes-13-00557-t004:** Specifications of the CO_2_ separation membrane module [36].

Specifications of Membrane Module			
Product	Model		Material
Hollow fibre	MCB-1512A		Polysulfone
Dimensions and Weight			
Length	Dimension		Weight
360 mm	55 mm		0.9 kg
Fibre Specifications			
No.	length	OD	ID
3800	280 mm	300 μm	160–180 μm
Operating Conditions			
Pressure	Temp. (Min/Max)	Relative Humidity	Residual oil
Max 10 bar	5/50 ℃	Less than 60%	≤0.01 $mg/m^{3}$

**Table 5 membranes-13-00557-t005:** Limits and steps of the study parameters.

Step No.	Feed Pressure (Bar)	Feed Temp. (°C)	CO_2_ in Feed Mol%
1	2.5	20	2
2	5	30	6
3	7.5	40	10

**Table 6 membranes-13-00557-t006:** The runs with the results obtained after conducting the experiments.

Run	Feed Pressure (Bar)	Feed Temp. (°C)	CO_2_ Mol% Feed	CH_4_ Mol% Feed	Permeate Flow Rate (L/Min)	CO_2_ Mol% Permeate	CH_4_ Mol% Permeate
1	2.5	20	2	98	0.334	23.013	76.982
2	2.5	20	2	98	0.3336	22.807	77.191
3	2.5	20	2	98	0.3329	22.43	77.568
4	2.5	30	6	94	0.2984	25.308	74.691
5	2.5	30	6	94	0.301	24.704	75.294
6	2.5	30	6	94	0.299	25.5304	74.467
7	2.5	40	10	90	0.2798	28.675	71.324
8	2.5	40	10	90	0.2761	27.324	72.673
9	2.5	40	10	90	0.2775	28.3201	71.678
10	5	20	2	98	0.612245	31.054	68.943
11	5	20	2	98	0.623	30.876	69.123
12	5	20	2	98	0.6204	31.342	68.655
13	5	30	6	94	0.5307	33.05	66.947
14	5	30	6	94	0.549	33.245	66.753
15	5	30	6	94	0.533	32.991	67.005
16	5	40	10	90	0.4558	35.343	64.654
17	5	40	10	90	0.459	34.673	65.326
18	5	40	10	90	0.4502	35.457	64.541
19	7.5	20	6	94	0.846	37.325	62.672
20	7.5	20	6	94	0.842	37.01	62.987
21	7.5	20	6	94	0.839	37.123	62.872
22	7.5	30	10	90	0.811	39.861	60.136
23	7.5	30	10	90	0.806	39.918	60.079
24	7.5	30	10	90	0.813	40.023	59.973
25	7.5	40	2	98	0.661	34.87	65.128
26	7.5	40	2	98	0.669	35.674	64.323
27	7.5	40	2	98	0.6605	34.9108	65.088

**Table 7 membranes-13-00557-t007:** Comparison of the mathematical model results with the CFD model.

$P_{F}$ Bar	$T_{F}$ K	$X_{F}$ Mol/Mol	Flux. Exp $Mol/m^{2}\cdot S$	Flux. Com $Mol/m^{2}\cdot S$	Relative Error	Recovery %CO_2_
2.5	293	2	7.31 × 10^−9^	6.93 × 10^−9^	5.26	9
2.5	303	6	2.07 × 10^−8^	2.00 × 10^−8^	3.31	18.65
2.5	313	10	1.04 × 10^−6^	9.56 × 10^−7^	7.73	19.1
5	293	2	1.42 × 10^−8^	1.38 × 10^−8^	2.63	19.2
5	303	6	2.24 × 10^−6^	2.01 × 10^−6^	5.94	21.4
5	313	10	8.96 × 10^−6^	8.57 × 10^−6^	4.36	23.4
7.5	293	6	9.49 × 10^−6^	9.22 × 10^−6^	2.83	28.3
7.5	303	10	2.39 × 10^−5^	2.31 × 10^−5^	3.33	30.3
7.5	313	2	2.18 × 10^−8^	2.17 × 10^−8^	0.391	20.6

**Table 8 membranes-13-00557-t008:** The concentrations of carbon dioxide in the sorption and desorption side in the membrane.

Length of Fiber m	Conc. of CO_2_ at *r*_1_ $Mol/m^{3}$	Conc. of CO_2_ at *r*_2_ $Mol/m^{3}$
0	8.8605	0
0.014	9.3512	7.1083
0.028	9.4065	8.4672
0.042	9.4901	9.089

## Data Availability

Not applicable.

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
