# Peer review of "Modeling and Optimal Operating Conditions of Hollow Fiber Membrane for CO2/CH4 Separation"

_membranes, 2023, doi:10.3390/membranes13060557_

Round 1

Reviewer 1 Report

The paper is for a standard gas separation module design. Nothing is particularly wrong. I recommend the publication after answering the following questions and comments.

In equation 14, what is AI?

In equation 14, what are P, PO and PI?

In equation 20, aren’t CCO2,T and CCH4,T CCO2,t and CCH4,t, respectively?

In equation 21, isn’t nN=0 Sigma Ni =N? And Isn’t N = -Di,t… Ni = -Di,t… Please let me know what n and N are.

In equation 24, isn’t N = -Di,t… Ni = -Di,t…?

In equations 28, 30 and 32, the same question as in equation 24.

In equation 31, the same question as in equation 21.

In equation 36 and 38, the same question as in equation 24.

In equation 37, the same question as equation 31.

The CO2 permeability of polysulfone is known to be about 4 x 10^-10 cm3 (STP) cm/cm2 s cmHg. The dense layer thickness of your hollow fiber is 200 x 10^-4 cm (very thick!). Then, the permeance is

4 x 10^-10/200 x 10^-4= 2 x 10^-8 cm3 (STP)/cm2 s cmHg.

 The hollow fiber has outer diameter of 300 x10^-4 cm and length of 28 cm. Your module contains 3800 fibers.

Then the permeate flow rate at 2.5 bar (ca 2.5 x 76 cmHg = 190 cmHg) will be

2 X10^-8 x 3.1416 x 300 x 10^-4 x 28 x 3800 x 190 = 3.81 x 10^-2 cm3/s

3.81x10^-2 x 60 = 2.29 cm3/min = 0.00229 L/min

This calculation is for pure CO2. When CO2 mole fraction is only 2 %, the flux should be even lower.

You are reporting 0.33 L/min in you table 6. Why can it be so high?

What does scale in Table 3 mean? Did you use 300 modules?

Reviewer 2 Report

Modeling and Optimal Operating Conditions of Hollow Fiber Mem- 2
brane for CO2/CH4 Separation

Ø  The literature survey is not up-to-date. The analysis should be supported with latest studied on current topic.

Ø  Explanation of each physical quantities appeared in Eqs. (1-3) needed.

Ø  The sentence, “After combining the three equations to delete the inter-mediate concentration..” is not cleared. Delete?

Ø  Proper commas/full stop needed after all equations.

Ø  The novelty and motivations of current analysis are not convincing. Different aspects of work should be explained comprehensively.

Ø  Discuss some limiting constraints of current  model.

Ø  How penetrating components into the membrane is linked to the volumetric fraction? More explanation needed.

Ø  Present some novel applications of current research in the last paragraph of abstract.

Ø  Include more physical impact of problem in discussion section.

Ø  Check English language again, For instance; On line 187, “Sherwood, Schmidt, and Reynolds numbers..”……” Equations 39-50 represents……”

Ø  The sentence on line 208 to 211 is very long and confusing.

Ø  On line 263, the word “their” is not appropriate.

Ø  Some physical explanation of boundary conditions (41-44) and (47-50) needed.

Ø  Include some source to Navier-Stockes equations needed.

Ø  Authors claimed that “In all the experiments, the feed flow rate was adjusted at 3.5084 L/min”. How this range is defined?

Ø  Conclusion section should be written in built form and modified.

Ø  One following study is relevant to current work

Photo-catalytic pretreatment of biomass for anaerobic digestion using visible light and Nickle oxide (NiOx) Nanoparticles prepared by sol gel method, Renewable Energy, Volume 154, (2020), pp. 128-135

Some typo errors should be corrected. 

Round 2

Reviewer 2 Report

Modeling and Optimal Operating Conditions of Hollow Fiber Membrane for CO2/CH4 Separation

The contest of work are still poor. No recommendations have been done. Authors should modify again. I repeat my comments again:

1.      Explanation of each physical quantities appeared in Eqs. (1-3) needed.

This point still need to be address. No description of physical quantities is presented. With out explanation of each physical quantity in model, how reader can understand the problem.

2.      Check the author response of comment (3). “Answer: What is meant is that it does not appear in the final equation and is disposed of.” I fails to understand which they done and even asked.

3.      Proper commas/full stop needed after all equations.

Authors responded that this has been done which is completely wrong. See eqs. (1-6). Where?

4.      How penetrating components into the membrane is linked to the volumetric fraction? More explanation needed. This comments is no addressed.

5.      Present some novel applications of current research in the last paragraph of abstract.

Author response: N/A

I am strange to see this response which is rejected. Again address this point clearly.

6.      Include more physical impact of problem in discussion section. This point again need to justify.

7.      The sentence on line 208 to 211 is very long and confusing.

Author response: N/A

This response is also rejected and need to modify.

8.      Conclusion section should be written in built form and modified.

Author response: N/A

In conclusion, the contest of work are still extremely poor and cannot be recommended. However, one more chance can be given to authors to modify the work.

English language is still exteremly poor. 

Author Response

We look forward to hearing from you regarding our submission and responding to any further questions and comments you may have.

Sincerely,

Round 3

Reviewer 2 Report

Authors have not highlighted the modifications which have been done. Please highlight the changes for assessment. 

Check some typo errors again. 

Round 4

Reviewer 2 Report

accept the current form

accept the current form